# Vaccine Therapy in Non-Small Cell Lung Cancer

**DOI:** 10.3390/vaccines10050740

**Published:** 2022-05-09

**Authors:** Miguel García-Pardo, Teresa Gorria, Ines Malenica, Stéphanie Corgnac, Cristina Teixidó, Laura Mezquita

**Affiliations:** 1Princess Margaret Cancer Centre, Toronto, ON M5G 2C1, Canada; miguel.garcia@uhn.ca; 2Medical Oncology Department, Hospital Clínic de Barcelona, 08036 Barcelona, Spain; tgorria@clinic.cat; 3Laboratory of Hepatobiliary Immunopathology, IRCCS Humanitas Research Hospital, Via Manzoni 56, 20089 Milan, Italy; ines.malenica@humanitasresearch.it; 4INSERM UMR 1186, Integrative Tumor Immunology and Immunotherapy, Gustave Roussy, Faculté de Médecine, Université Paris-Saclay, 94805 Villejuif, France; stephanie.corgnac@gustaveroussy.fr; 5Department of Pathology, Hospital Clínic of Barcelona, University of Barcelona, Villarroel 170, 08036 Barcelona, Spain; teixido@clinic.cat; 6Laboratory of Translational Genomics and Targeted Therapies in Solid Tumors, IDIBAPS, 08036 Barcelona, Spain; 7Department of Medicine, University of Barcelona, 08036 Barcelona, Spain

**Keywords:** therapeutic cancer vaccine, neoantigen, non-small cell lung cancer, immune checkpoint inhibitor, tumor antigen

## Abstract

Immunotherapy using immune checkpoint modulators has revolutionized the oncology field, emerging as a new standard of care for multiple indications, including non-small cell lung cancer (NSCLC). However, prognosis for patients with lung cancer is still poor. Although immunotherapy is highly effective in some cases, not all patients experience significant or durable responses, and further strategies are needed to improve outcomes. Therapeutic cancer vaccines are designed to exploit the body’s immune system to activate long-lasting memory against tumor cells that ensure tumor regression, with minimal toxicity. A unique feature of cancer vaccines lies in their complementary approach to boost antitumor immunity that could potentially act synergistically with immune checkpoint inhibitors (ICIs). However, single-line immunization against tumor epitopes with vaccine-based therapeutics has been disappointingly unsuccessful, to date, in lung cancer. The high level of success of several recent vaccines against SARS-CoV-2 has highlighted the evolving advances in science and technology in the vaccines field, raising hope that this strategy can be successfully applied to cancer treatments. In this review, we describe the biology behind the cancer vaccines, and discuss current evidence for the different types of therapeutic cancer vaccines in NSCLC, including their mechanisms of action, current clinical development, and future strategies.

## 1. Introduction

Lung cancer is currently the leading cause of cancer death worldwide [1]. Despite many recent advances, there are more than 2 million new lung cancer cases every year, with an increasing number in non-smokers and women [1,2]. Non-small cell lung cancer (NSCLC) represents about 85% of newly diagnosed lung cancer cases and the majority of patients have advanced stage disease at diagnosis [3].

Over the last decade, major improvements have been achieved in treatments for NSCLC, thanks to the development of targeted therapies and immunotherapy [4]. Immune checkpoint inhibitors (ICIs) have become the cornerstone of NSCLC treatment in the advanced setting [5,6,7,8], with some patients achieving responses with durations and prolonged survival previously unheard of in the metastatic NSCLC setting [9]. However, not all patients benefit from a long-term response with ICIs [10,11]. The infiltration by T-cells capable of detecting and killing cancer cells seems to be the key to response, with data showing that ICIs are not as effective in ‘cold tumors’, which are characterized by a lack of T-cell infiltration [12]. Further strategies are needed to improve patient outcomes, extending the benefit to more patients and avoiding unnecessary exposure for patients who are unlikely to benefit. Additional strategies with immunotherapy, including cancer vaccines, designed to restore the cancer-immunity cycle [13] are under investigation in solid tumors and NSCLC, however results thus far have been disappointing.

## 2. The Anticancer Immune Response

The spectacular therapeutic impact of recent vaccine clinical development in the management of SARS-CoV-2 (COVID-19) has been seen globally. However, successful immunization in other fields, including against tumor epitopes via vaccine-based therapeutics—the subject of research for many years—is lagging behind, and has not resulted in the hoped-for outcome in lung cancer. The body’s natural specific T-cell response against tumor cells relies on several steps. The first step is the tumor antigen encountered by antigen-presenting cells (APCs) such as dendritic cells (DCs). The antigen-loaded APCs ultimately reach the draining lymph nodes, where T cell activation or priming occurs, giving rise to antigen-specific T-cells. DCs present the antigens on MHC class I and class II molecules to CD8+ and CD4+ T-cells, respectively. The generation of tumor-specific T cell responses occurs via costimulatory signals that allow for the generation and expansion of activated tumor-specific CD4+ and CD8+ T-cells. This step is crucial, as the nature of the immune response depends on the critical balance between activated conventional CD4+ effector cells and CD4+CD25^high^. T effector cells and CD4+CD25^high^ T regulatory cells express master transcription factor FoxP3, with the latter having a detrimental effect on immune response in multiple solid cancers [14,15]. Finally, the activated and expanded T-cells travel to the tumor site, infiltrate the tumor bed, and recognize and kill cancer cells. Killing of the tumor cells releases additional antigens, increasing the diversity and breadth of the immune response [13].

Cancer vaccines are designed to select suitable antigen targets to induce new antigen-specific T cell responses against tumor cells, to amplify pre-existing responses, and to increase the breadth and diversity of the tumor-specific T cell response [16]. The main challenge is that, unlike SARS-CoV-2 or other viruses, tumor antigens arising from abundant, but non-mutated, proteins are often recognized by the immune system as ‘self’, limiting an immune response and clinical benefit. In patients with cancer, the natural specific T-cell response against foreign antigens does not function optimally and the cycle is altered at different steps. Additionally, there is often an immunosuppressive tumor microenvironment suppressing tumor infiltration by effector cells [17].

As a result, cancer vaccine research is evolving along new avenues to stimulate the immune system, such as with tumor-specific antigens (TSAs), not expressed by healthy tissue, as well as combination strategies to induce deeper immune responses [18].

In addition to tumor antigens, the development of a successful cancer vaccine depends on another two key components: immune adjuvant platforms, and delivery vehicles. The identification of neoantigens and novel platforms are major advances in this field, and are reviewed in detail here.

## 3. Cancer Vaccines: TAAs and Antigen Selection

### 3.1. Prophylactic Cancer Vaccines

Cancer vaccines can be explored in the prophylactic or the therapeutic setting. Prophylactic vaccines are used to prevent infection by oncogenic viruses, while therapeutic vaccines aim to boost the immune system to eradicate neoplastic cells and revolve around tumor antigens.

Viral antigen-based vaccines against hepatitis B virus (HBV) and human papillomavirus (HPV) represent the paradigm for prophylactic cancer vaccines, with both used to prevent infections caused by known oncogenic viruses [18]. Vaccination against cancer-promoting viral infections such as HBV (a risk factor for hepatocellular cancer) and HPV (which promotes the onset of cervical cancer) have demonstrated a substantial reduction in cancer incidence [19,20]. However, the development of vaccines for cancers with no known viral etiology is much more challenging [21].

A prophylactic vaccine is not yet considered to be a promising option for lung cancer since oncogenic viruses involved in NSCLC pathogenesis have not been identified. Choosing the right tumor antigen, expressed only by tumor tissue, to promote an immune response in the host remains challenging, as well-defined tumor antigens are usually expressed by normal tissues [22].

### 3.2. Antigen Selection

Tumor antigens can be grouped in two main categories: tumor-associated antigens (TAAs), which are self-antigens abnormally expressed in cancer cells and, therefore, also expressed in normal cells; and TSAs or neoantigens, which are encoded by tumor-specific somatic mutations and, therefore, only expressed by neoplastic cells [16].

The first and most crucial step in cancer vaccine development is the choice of an antigen, which should ideally satisfy several conditions: that it is expressed uniquely by cancer cells and not by normal cells, that it is homogenously expressed across all cancer cells to avoid cancer escape and metastasis development, that it is essential for cancer cell survival, and that it is highly immunogenic [23].

In the past, TAAs have been widely investigated but have shown limited clinical benefit, likely due to preexisting central tolerance to self-antigens [16,23]. Immune responses to TAAs have been reported, but phase III trials of vaccines targeting TAAs in NSCLC (MAGE-A3, MUC-1) have been negative thus far [24].

Unlike TAAs, neoantigens are tumor and often patient-specific; they arise from molecular alterations and are less likely to be subject to central tolerance [25]. Solid tumors carry molecular alterations that result in aberrant proteins, which can be identified by the host immune system. Genomic instability is present in NSCLC, leading to the generation of neoantigens. Nevertheless, not all types of genomic instability have the same impact when viewed in the context of neoantigen generation or response to immunotherapy [26].

Additionally, it is not feasible to determine neoantigen load in daily practice. Tumor mutation burden (TMB), defined as the total number of somatic/acquired mutations per coding area of a tumor genome (Mut/Mb), has failed to consistently predict response to ICIs thus far [27,28].

With the ongoing advances in next-generation sequencing technology and bioinformatics, neoantigen identification for cancer vaccines development is becoming increasingly feasible [29]. KRAS-mutated and pan-wild-type NSCLC, both typically associated with smoking history, are associated with a higher number of predicted neoantigens [30], although only a small proportion of predicted neoantigens leads to detectable spontaneous immune responses [29].

## 4. Platforms Developed for Vaccine Therapy

The four main types of vaccine platforms encoding TAAs that have been developed for cancer therapy are cell-based, protein/peptide-based vaccines, viral vectors, and gene-based vaccines (DNA or RNA). They are reviewed here, with a focus on NSCLC, and are summarized in Table 1.

### 4.1. Cell-Based Vaccines

Cell-based vaccines are designed using dead or living tumor cells that are able to induce an immune response in the host, with the aim of protecting the host from future disease or infection. Strategies employing this approach include the use of allogenic tumor cell lines, irradiated autologous tumor cells, or autologous tumor lysates [40]. Early studies using vaccines with irradiated whole tumor cells alone were unsuccessful; however, using several cytokines such as interleukin-2 (IL-2) or granulocyte-macrophage colony stimulating factor (GM-CSF) increased the efficacy of whole cell vaccines [41,42].

The first FDA approved cancer vaccine was Sipuleucel-T for metastatic castration resistant prostate cancer. Sipuleucel-T is an autologous dendritic cell-based vaccine loaded with a prostatic acid phosphatase (PAP) antigen fused to GM-CSF. Despite a small but statistically significant survival benefit demonstrated in a phase III trial [43], several limitations such as the cost and complexity of its production, barriers to administration, and the problematic cross-over [44], have prevented its widespread use.

Cell-based vaccines have been investigated in NSCLC, but with disappointing results. The GVAX vaccine is a whole tumor cell vaccine, genetically modified to secrete GM-CSF. A platform composed of autologous tumor cells mixed with an allogenic GM-CSF secreting cell line was developed. Safety and efficacy were explored in a phase I/II trial; however, objective tumor responses were not seen [31]. Efficacy in other tumor types has also been very limited [45,46].

Belagenpumatucel-L is an allogeneic whole tumor cell vaccine containing four NSCLC cell lines transfected with a human transforming growth factor (TGF)-β2-antisense vector designated pCHEK/HBA2. Phase II trials initially demonstrated the safety of this cell-based vaccine in patients with NSCLC, with encouraging efficacy [47,48]. A phase III trial was conducted in advanced NSCLC. Maintenance belagenpumatucel-L after platinum-based chemotherapy was compared with placebo; however, there were no differences in overall or progression-free survival between belagenpumatucel-L and placebo [32].

1650-G is an allogenic cellular vaccine comprising a lethally irradiated tumor cell allogenic line and GM-CSF. A pilot study conducted by Hirschowitz et al., in 11 patients with NSCLC confirmed safety and demonstrated an immunological response in 6 patients [33]. Clinical efficacy has not yet been proven in larger, randomized clinical trials.

### 4.2. Peptide/Protein-Specific Vaccines

Peptide-based vaccines mimic the epitopes of the antigen that triggers an anticancer immune response. Early studies with single-antigen based short peptides have not shown robust immune responses. In the pre-ICI era, the gp100 peptide vaccine in addition to IL-2 was associated with a higher response rate and longer progression-free survival compared to IL-2 alone in patients with advanced melanoma [49]. Several peptide/protein specific vaccines have been investigated in NSCLC.

The MAGE-A3 cancer vaccine has been explored in the adjuvant setting, in patients with resected NSCLC. This vaccine combines recombinant MAGE-A3 protein (a TAA expressed in 30–50% of NSCLC) which is given with AS15, an immunostimulant. A randomized phase III clinical trial (MAGRIT, NCT00480025) was conducted in patients with fully resected stage IB-IIIA MAGE-A3-positive NSCLC. Patients were randomized to receive MAGE-A3 vaccine versus placebo. The outcome of this trial was negative and there were no differences in disease-free survival between groups [24]. MAGE-A3 immunotherapeutic use in NSCLC has consequently been terminated.

CIMAvax-EGF is an epidermal growth factor (EGF)-based vaccine, composed of human recombinant EGF coupled to a carrier protein, recombinant P64. A randomized phase II trial using CIMAvax-EGF in advanced NSCLC was developed in Cuba [50]. CIMAvax-EGF was administered as maintenance or second-line therapy versus best supportive care, and a trend towards a survival benefit was found [50,51]. A phase III trial was subsequently launched in patients with advanced NSCLC after first-line chemotherapy, who were randomized to CIMAvax-EGF or a control group, treated with best supportive care. Patients received treatment every two weeks for four doses (induction period) and then monthly. The vaccine was safe, induced anti-EGF antibodies, and decreased serum EGF concentrations. In the intention-to-treat population, median overall survival was 10.8 months in the vaccine arm versus 8.9 months in the control arm, which was not statistically significant (HR, 0.82; *p* = 0.1) [34]. CIMA-vax-EGF is approved in Cuba, Peru, and Venezuela for the treatment of stage IIIB-IV NSCLC after progression on first-line chemotherapy, and a combination of CIMAvax-EGF and anti-PD1 antibodies is being explored in several clinical trials.

Racotumomab-alum is an anti-idiotype vaccine targeting the NeuGcGM3 tumor-associated ganglioside. Neu glycolyl (NeuGc)-containing gangliosides are glycolipids present on NSCLC cells, but not on the cytoplasmic membrane of normal cells [52,53]. Several phase I trials using this vaccine in patients with advanced NSCLC demonstrated adequate safety and immunogenicity. A high antibody response to NeuGcGM3 was identified in vaccinated patients [54,55]. In 2014, a phase II/III trial was conducted. Patients with advanced NSCLC with at least stable disease after first-line chemotherapy were randomized to racotumomab-alum or placebo. Median overall survival was 8.2 months in the vaccine group and 6.8 months in the placebo group (HR, 0.63; 95% CI, 0.46–0.87; *p* = 0.004). Median progression-free survival in vaccinated patients was 5.3 months versus 3.9 months for placebo (HR, 0.73; 95% CI, 0.53–0.99; *p* = 0.039) [35]. Racotumomab is approved in Argentina and Cuba for advanced NSCLC after progression on first-line therapy.

Tecemotide (L-BLP25) is a MUC1 antigen-specific vaccine. Preclinical and early-phase studies showed adequate safety and immunogenicity, as well as preliminary efficacy signals [56,57]. A phase III trial of L-BLP25 after chemoradiation in patients with unresectable stage NSCLC was conducted (START). The study was negative, as there was no significant difference in overall survival with the administration of tecemotide after chemoradiotherapy compared with placebo [36].

The PReferentially expressed Antigen in Melanoma (PRAME) is a testis-selective cancer testis antigen (CTA) with restricted expression in somatic tissues and preferentially expressed in variety of cancers, including NSCLC, metastatic melanoma, breast cancer, and neuroblastoma [37]. The safety and tolerability of the adjuvant PRAME vaccine, consisting of recombinant PRAME plus proprietary immunostimulant AS15, was explored in a phase I dose escalation trial including 60 patients with surgically resected NSCLC; anti-PRAME humoral responses were demonstrated with no cancer regression, and the trial was stopped due to negative results [37].

### 4.3. Viral Vaccines

Viral vector-based vaccines are used to administer antigens in the host and provoke an immune response. In 2015, the FDA approved the first oncolytic virus vaccine talimogene laheparepvec (T-VEC) for advanced melanoma [58]. T-VEC is a type I herpes simplex virus, genetically altered to replicate in cancer cells, promote antigen loading of MHC class I molecules, and express GM-CSF to boost tumor-antigen presentation by DCs [59]. Following intratumoral injection, it provokes cell lysis, inducing antitumor responses not only at the injection site but also in distant metastases [60]. T-VEC is being investigated in different solid tumors alone or in combination of immunotherapy (NCT02509507). The intralesional administration of this therapy makes it a logistical challenge for lung cancer patients, whose lesions are often not easily accessible.

Another viral-based vaccine, PROSTVAC, used recombinant poxviruses expressing prostate-specific antigen in castration-resistant prostate cancer. However PROSTVAC was discontinued due to lack of efficacy, and this technology has not been extended to other tumor types [61].

In advanced NSCLC, viral-vector based vaccines targeting TAAs have been investigated with disappointing results. TG4010 is a recombinant viral-based vaccine targeting MUC, a TAA overexpressed in many solid tumors. Phase II clinical trials of TG4010 in combination with chemotherapy demonstrated encouraging efficacy results and no safety signals [38]. TG4010 therapy has been demonstrated to correlate with T cell responses against MUC1, and broader CD8 responses were also seen against non-targeted TAA, likely due to the induction of epitope spreading [62]. TG4010 is currently being investigated in combination with nivolumab in the second-line setting (NCT02823990).

A phase I/II trial of the MG1-MAGEA3 vaccine constructed with an adenovirus expressing MAGE-A3 in combination with pembrolizumab is ongoing (NCT02879760). This exploration is based on preclinical models showing superior tumor clearance and survival, and induction of CD8+ T-cell infiltration turning ‘cold’ tumors into ‘hot’ tumors [63].

LV305 is a lentivirus-based vaccine designed to selectively transduce DCs in vivo through the DC-SIGN receptor. LV305 induces expression of NY-ESO-1 cancer testis antigen in DCs, to promote immune responses against NY-ESO-1-expressing tumors. A phase I study including patients with advanced sarcoma, ovarian cancer, melanoma, and lung cancer showed a favorable safety profile, induction of antigen-specific responses, and potential clinical activity [39]. Ongoing research is exploring the role of T cell receptor engineered-T cells (TCR-T cells) to specifically recognize NY-ESO-1 antigen with increased avidity [64].

### 4.4. DNA/RNA Vaccines

DNA vaccines use plasmids to deliver genes encoding tumor antigens, stimulating the adaptative immune response towards cancer cells. The DNA vaccine pVAX1-MAGEA3-sPD1 is based on the MAGE-A3 antigen, enhanced with soluble PD-1 (sPD1). A series of DNA plasmids encoding MAGE-A3, and the extracellular domain of murine PD-1 (sPD1) was developed and showed immunogenicity and tumor growth inhibition in mice [65].

RNA vaccines offer some advantages over DNA vaccines thanks to their mechanism of action. Unlike DNA, which has to be integrated into the DNA machinery in order to be expressed, RNA only needs to enter the cytoplasm for expression, limiting its oncogenic potential [66], and reinforcing the overall safety in a therapeutic context. Several mRNA vaccines platforms are under development in the cancer setting, and there has been renewed interest in mRNA technology as a result of the success of the SARS-CoV-2 vaccines [67].

## 5. Current Perspectives and Future Directions

### 5.1. Recent Vaccine Success against SARS-CoV-2

The extremely rapid development and high level of efficacy of several vaccines against SARS-CoV-2 [68,69] has highlighted the major advances in science and technology in the vaccines field. Although therapeutic cancer vaccines have shown very little benefit in clinical practice thus far, this recent progress provides promise that this strategy can be applied more widely, beyond the prevention of infectious diseases, such as for cancer treatment.

Several technology platforms have been explored for the development of SARS-CoV-2 vaccines, including whole virus vaccines [70], nucleic acid vaccines [68,69], protein-subunit based vaccines [71], and viral vector-based vaccines [72,73]. To date, the most successful strategies have been the viral vector-based and nucleic acid vaccines. Two mRNA vaccines, the mRNA-1273 vaccine (Moderna) and the BNT126B1 (BioNTech & Pfizer) have made the most progress towards wining the vaccine race in 2020. They are the preferred vaccines in much of Europe, America, and Australasia [74] for the prevention of detrimental patient outcomes following infection with the SARS-CoV-2 virus, due to their high efficacy and superior safety profile compared to other vaccines authorized to date. It is hypothesized that the effective delivery method of personalized cancer, based on well-known mRNA vaccine technology, could optimize immune cell responses toward immune-escaped lung tumors [75].

### 5.2. mRNA Vaccines

As some of the mechanisms of acquired resistance to immunotherapies correlate with downregulation of MHC-I and defects in antigen-presenting machinery, the signal-sequence-derived peptides and their carrier proteins, such as ppCT [75] are attractive candidates for specific therapeutic cancer vaccines. mRNA vaccines represent one of the most promising fields, and the translation of 30 years of research into novel mRNA vaccines in record time [76] has ensured a dramatic reduction in fatal outcomes associated with SARS-CoV-2 infection. This has changed the way we consider vaccine preparation and clinical use. Extending this knowledge to benefit the field of cancer vaccines is an obvious next step, and enthusiasm for mRNA vaccines in cancer has been renewed in the post-pandemic era. The first phase II trial of an mRNA-based vaccine in patients with advanced melanoma was recently initiated (NCT04526899), based on a favorable safety profile as well as durable objective responses observed in patients who had progressed following prior check-point blockade targeting PD-1 [77]. Combination strategies are also being explored. Haanen et al. presented promising interim results of a phase I clinical trial in solid malignancies combining claudin-CAR T cells with a CLDN6-encoding mRNA vaccine (CARVac). The mRNA vaccine favored the expansion of the CAR T cells, and this novel therapy showed promising early results in terms of response and safety [78,79].

mRNA vaccines are also under investigation in NSCLC. CV9201 and CV9202 are mRNA-based cancer vaccines containing sequence-optimized mRNAs encoding different cancer antigens to facilitate antigen expression and activation of the immune system, inducing an adaptive cellular and humoral immune response [80,81,82]. A previous phase I/IIa study evaluating the RNActive^®^-derived CV9201 cancer vaccine in 46 patients with advanced NSCLC demonstrated an adequate safety profile. Immune responses against all five encoded antigens (NY-ESO-1, MAGE-C1, MAGE-C2, Survivin, and TPBG) were reported [82]. CV92102, a similar RNA-based vaccine encoding 6 TAA (NY-ESO-1, MAGE-C1, MAGE-C2, Survivin, 5T4, and MUC) was also well tolerated, and antigen-specific immune responses were detected in a phase Ib study enrolling 26 patients with stage IV NSCLC [81].

### 5.3. Neoantigens and Personalized Cancer Vaccines

A deeper understanding of the basic mechanisms of T cell recognition of neoantigens, and computational approaches to discover somatic mutations and neoantigen prediction, will likely enable greater clinical benefit with future neoantigen-based immunotherapies [83]. Advances in the field will allow for personalized cancer vaccines based on each patient’s particular neoantigens, with the goal of inducing high-affinity immune T-cell responses. Here, we provide some examples of promising research in the field.

The neoantigen peptide-based vaccine, NEO-PV-01, has recently been evaluated in a phase Ib trial in combination with nivolumab (an anti-PD-1 ICI) in patients with NSCLC, bladder cancer, or melanoma. It showed an acceptable safety profile and a response rate of 39% in the 18 patients treated in the NSCLC cohort [84].

Vx-001 is a vaccine based on optimized cryptic peptides, a new family of tumor antigens which are derived from universal tumor antigens but are identified by the immune system as neoantigens or ‘non-self’, being strongly immunogenic. Vx-001 targets TERT (TElomerase Reverse Transcriptase) and has been tested in a phase I/II clinical study in patients with different solid tumors, mainly NSCLC. However, a phase II clinical trial found no differences in overall survival, disease control rate, or time to treatment failure [85]. Several other early-phase neoantigen vaccine trials are also currently enrolling patients with lung cancer.

RO7198457 is an RNA-Lipoplex Individualized Neoantigen-Specific Immunotherapy (iNeST), with up to 20 patient-specific neoantigens. It is manufactured on a per-patient basis, and it aims to induce T cell responses against neoantigens. A first-in-human phase Ia study of RO7198457 was conducted in patients with locally advanced or metastatic solid tumors, and demonstrated a manageable safety profile and strong neoantigen-specific immune responses [86]. A combination study with ICIs is ongoing (NCT03289962).

### 5.4. Oncogenic Driver Vaccines in NSCLC

KRAS mutation is a genetic driver of multiple cancers, including NSCLC. Neoantigens encoded by KRAS mutations are tumor-specific and highly immunogenic [87]. KRAS mutation-based cancer vaccines have shown encouraging preclinical results, however efficacy may be limited by an immunosuppressive tumor microenvironment. Combination strategies are under exploration in several clinical trials (NCT05202561, NCT05254184, and NCT04117087).

ALK rearrangement is found in approximately 5–6% of NSCLC. In preclinical studies by Voena et al., an ALK vaccine containing a DNA plasmid coding the intracytoplasmic domain of ALK was shown to be effective in a mouse model of ALK-rearranged NSCLC by inducing a tumor-specific cytotoxic response [88]. A phase I trial to test this approach in humans is planned [89,90].

## 6. Challenges and Limitations

Therapeutic cancer vaccines aim to establish the body’s long-lasting immunological memory against tumor cells, thereby leading to effective tumor regression while minimizing non-specific or adverse events [91]. While an attractive potential complement to ICIs due to their safety, specificity, and long-lasting response following activation of immune memory, therapeutic cancer vaccine monotherapy has been disappointingly unsuccessful to date. Today, the clinical focus is on developing efficient therapeutic cancer vaccines that are able to promote an effective and durable T cell response to specific tumor antigens [91]. Nevertheless, there are four major challenges that therapeutic cancer vaccines must override: low immunogenicity, the immunosuppressive tumor microenvironment, established disease burden, and inefficient long-term memory generation [22,23].

The main reason for cancer vaccine failure is the complexity of identifying specific target tumor antigens shared by multiple tumor types, and that are also unique or overexpressed only by the tumor cells compared to normal tissue. Most tumor antigens incorporated in cancer vaccines have been TAAs or non-mutated overexpressed self-antigens; however, the high-affinity T cells recognizing self-antigens are eliminated during development by the immune system’s tolerance mechanisms, limiting the potential vaccine efficacy. Therefore, T cells activated by such vaccines would be those with low-affinity T cell receptors (TCR), which are unable to stimulate an effective antitumor response [92], and additional mechanisms are needed.

Moreover, cancer cells can develop mechanisms for preventing immune cells from infiltrating the tumor microenvironment (TME) despite adequate immunogenicity [93], such as by secreting immunosuppressive chemokines, by the induction of the loss of MHC antigen expression, and by recruiting higher number of immunosuppresive cells in the TME [94].

Cancer cells can also downregulate the expression of target antigens, MHC or co-stimulatory molecules leading to failure of T-cell recognition [95]. Lung cancer cells have been shown to produce a variety of immunosuppressive molecules including TGF-beta, prostaglandin E2, IL-10, and cyclooxygenase-2 that can affect dendritic cell processing and presentation, as well as the effector functions of cytotoxic T lymphocytes [96,97]. Several strategies have been investigated to overcome immunosuppressive mechanisms of the TME and counteract tumor escape, including improving antigen selection, refining immunotherapy delivery platforms, and combination therapies [25].

## 7. Conclusions

The advent of cancer immunotherapy using ICIs has revolutionized clinical treatment of multiple solid cancers, including NSCLC. However, further strategies are needed as less than one third of patients with non-oncogene addicted advanced NSCLC benefit from ICIs long-term [98]. Cancer vaccines can potentially induce new specific T cell responses against tumor antigens and amplify existing responses. However, therapeutic cancer vaccines have demonstrated very limited clinical benefit in NSCLC thus far. The recent success of mRNA-based vaccines against SARS-CoV-2 has highlighted the evolving vaccines field, raising renewed hope that this strategy can be successfully applied for cancer treatment. We anticipate exciting new results from the ongoing clinical trials with mRNA vaccines, while the identification of more potent neoantigens through artificial intelligence and epitope prediction algorithms, and the development of new vaccine technologies and combination strategies with ICIs, represent additional avenues to explore for the treatment of lung cancer and other solid tumors.

## Figures and Tables

**Table 1 vaccines-10-00740-t001:** Clinical trials of NSCLC vaccines encoding TAAs.

Name	Formulation	TAA	Phase	Number of Patients Enrolled	Disease Stage	Endpoints	Results	NCT Identification	Reference
**Cell-based**									
GVAX	Cell-based	Autologous tumor cells mixed with an allogeneic GM-CSF-secreting cell line	I/II	86	IV	Safety and vaccine manufacturing feasibility	Positive	NCT00074295	Nemunaitis et al., 2006 [31]
Belagenpumatucel-L	Cell-based	4 TGF-β2-antisense gene-modified, irradiated, allogeneic NSCLC cell lines	III	532	IIIA-B/IV	OS	Negative	NCT00676507	Giaccone et al., 2015 [32]
1650-G	Cell-based	Allogeneic NSCLC cell line 1650 + GM-CSF	II	12	I-IIB (adjuvant)	Measurable immunologic response to vaccine	Positive	NCT00654030	Hirschowitz et al., 2011 [33]
**Peptide-based**									
MAGE-A3	Peptide-based	MAGE-A3	III	2312	IB-IIA (adjuvant)	PFS	Negative	NCT00480025	Vansteenkiste et al., 2016 [24]
CIMAvax-EGF	Peptide-based	Epidermal Growth Factor	III	579	IIIB-IV	OS	Negative	-	Rodriguez et al., 2016 [34]
Racotumomab-alum	Peptide-based	NeuGcGM3	III	1082	IIIA-IV	OS	Positive	NCT01460472	Alfonso et al., 2014 [35]
Tecemotide (L-BLP25)	Peptide-based	MUC1	III	1513	III	OS	Negative	NCT00409188	Butts et al., 2014 [36]
PRAME	Peptide-based	PRAME	I	60	IB, II, IIIA	Dose limiting toxicity and humoral immune response	Negative	NCT01159964	Pujol et al., 2016 [37]
**Virus-based**									
TG4010	Virus-based	MUC1	II	65	III/IV	Tumor Response	Negative	NCT00415818	Ramlau et al., 2008 [38]
LV305	Virus-based	NY-ESO-1	I	47	III/IV	Safety and tolerability	Positive	NCT02122861	Somaiah et al., 2019 [39]

## Data Availability

Not applicable.

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
