# Peer review of "Vaccine Therapy in Non-Small Cell Lung Cancer"

_vaccines, 2022, doi:10.3390/vaccines10050740_

Round 1
Reviewer 1 Report
In this article, the authors aim to discuss the biology and MOA of various kinds of cancer vaccines relevant in the field of NSCLC treatment and provide future perspectives about the trajectory of vaccine therapy in NSCLC. Overall, this is a very succinct review of the cancer vaccine field in NSCLC. It is well articulated and eloquently communicated. Some minor revisions are required before this article can be accepted for publication. Please see my comments below.
Several mRNA vaccines platforms are under development in the cancer setting, and there has been renewed interest in mRNA technology as a result of the success of the SARS-CoV-2 vaccines [65].
-In this context, I strongly urge the authors to add the recent mRNA vaccine study from Biontech showing promising interim results in a clinical trial in solid malignancies
https://www.aacr.org/about-the-aacr/newsroom/news-releases/new-car-t-cell-therapy-for-solid-tumors-was-safe-and-showed-early-efficacy/
https://www.science.org/doi/10.1126/science.aay5967?url_ver=Z39.88-2003&rfr_id=ori:rid:crossref.org&rfr_dat=cr_pub%20%200pubmed
Future perspectives
mRNA vaccine
-It might be beneficial to separate the post-pandemic initiatives from the pre-pandemic ones. Enthusiasm for mRNA vaccines has been renewed in the post-pandemic era. However, some of the citations in this section still date to pre-pandemic times.
-What is not clear is from which time point the authors are calling the research/trials a ‘ future’ perspective. A relatively clear distinction of timeline representing the past and current efforts and future perspectives would be helpful for this review.
-Besides the three challenges highlighted in vaccine development, long-term memory response is also a challenge in developing anti-cancer vaccines resulting in relapse.
Conclusion
however further strategies are needed as the majority of patients do not benefit from this therapy in long-term
-This is a relatively ambiguous statement as the authors do not clarify/discuss some of the strategies that might be beneficial to potent vaccine development.
Author Response
Point 1: In this article, the authors aim to discuss the biology and MOA of various kinds of cancer vaccines relevant in the field of NSCLC treatment and provide future perspectives about the trajectory of vaccine therapy in NSCLC. Overall, this is a very succinct review of the cancer vaccine field in NSCLC. It is well articulated and eloquently communicated. Some minor revisions are required before this article can be accepted for publication. Please see my comments below.
Response 1: We thank the reviewer for the comment.
Point 2: Several mRNA vaccines platforms are under development in the cancer setting, and there has been renewed interest in mRNA technology as a result of the success of the SARS-CoV-2 vaccines [65].
-In this context, I strongly urge the authors to add the recent mRNA vaccine study from Biontech showing promising interim results in a clinical trial in solid malignancies
https://www.aacr.org/about-the-aacr/newsroom/news-releases/new-car-t-cell-therapy-for-solid-tumors-was-safe-and-showed-early-efficacy/
https://www.science.org/doi/10.1126/science.aay5967?url_ver=Z39.88-2003&rfr_id=ori:rid:crossref.org&rfr_dat=cr_pub%20%200pubmed
Response 2: Many thanks for your comments. We added a comment about the BNT211 phase I trial from Biontech presented in AACR, and we have added the two references suggested
Point 3: It might be beneficial to separate the post-pandemic initiatives from the pre-pandemic ones. Enthusiasm for mRNA vaccines has been renewed in the post-pandemic era. However, some of the citations in this section still date to pre-pandemic times.
Response 3: Thank you for the suggestion. We have added a sentence noting that the enthusiasm for mRNA vaccines has been renewed in the post-pandemic era
Point 4: What is not clear is from which time point the authors are calling the research/trials a ‘ future’ perspective. A relatively clear distinction of timeline representing the past and current efforts and future perspectives would be helpful for this review.
Response 4: We thank the reviewer for the comment. We have reworded the title, “Current perspectives and future directions” instead of “Future perspectivesd”. In Conclusions, we discuss how new efforts such as combination therapies, the identification of more potent neoantigens, and mRNA vaccines are the new strategies under investigation.
Point 5: Besides the three challenges highlighted in vaccine development, long-term memory response is also a challenge in developing anti-cancer vaccines resulting in relapse.
Response 5: Thank you for your comment. We have added “inefficient long-term memory generation” as a relevant challenge in developing cancer vaccines, and we have added a reference
Point 6: Conclusion
however further strategies are needed as the majority of patients do not benefit from this therapy in long-term
-This is a relatively ambiguous statement as the authors do not clarify/discuss some of the strategies that might be beneficial to potent vaccine development.
Response 6: We thank the reviewe for his comment. We have reworded the sentence. We highlight the fact that less than less than one third of the patients with non-oncogene addicted advanced NSCLC benefit long term from immune-check point inhibitors, and we discuss some of the strategies that might be beneficial to potent vaccine development including combination therapies, the identification of more potent neoantigens, and mRNA vaccines.
Reviewer 2 Report
Dear Editor and Authors,
It was a pleasure to evaluate this review article titled “Vaccine therapy in non-small cell lung cancer” by Dr. García-Pardo and colleagues in which they describe the progress made in the development of a lung cancer vaccine, present the biology and mechanisms of action behind them, discuss the current evidence for the different types of therapeutic NSCLC vaccines NSCLC and propose future strategies.
This is a well written and presented work. It is thorough and encompasses all available strategies and work while at the same time the authors have managed to remain clear and succinct not overburdening the reader with extraneous minutiae! The good use of tables is also contributory to this. In terms of language it is clear and simple with only a couple of minor expression mistakes seen.
I was quite intrigued by the whole presentation and as a physician quite hopeful that as the authors present it that with the advent of mRNA vaccines for SARS-Cov-2 we are a step closer in the development of a cancer vaccine for NSCLC. Thank you again.
Author Response
Point 1: Dear Editor and Authors,
It was a pleasure to evaluate this review article titled “Vaccine therapy in non-small cell lung cancer” by Dr. García-Pardo and colleagues in which they describe the progress made in the development of a lung cancer vaccine, present the biology and mechanisms of action behind them, discuss the current evidence for the different types of therapeutic NSCLC vaccines NSCLC and propose future strategies.
This is a well written and presented work. It is thorough and encompasses all available strategies and work while at the same time the authors have managed to remain clear and succinct not overburdening the reader with extraneous minutiae! The good use of tables is also contributory to this. In terms of language it is clear and simple with only a couple of minor expression mistakes seen.
I was quite intrigued by the whole presentation and as a physician quite hopeful that as the authors present it that with the advent of mRNA vaccines for SARS-Cov-2 we are a step closer in the development of a cancer vaccine for NSCLC. Thank you again.
Response 1: We thank the reviewer for the nice comment and recognition of our work. We agree we are a step closer in the development of an effective therapeutic cancer vaccine for NSCLC.
Reviewer 3 Report
Garcia-Pardo et al have written a brief overview on the potential for future vaccine development in NSCLC. They nicely compile the past attempts and outcomes from multiple clinical trials. This is a good, but brief review on this topic.
Major Points
Increased discussion on genomic stability in NSCLC and its role in generating neoantigens
Minor Points
Line 66 appears to have an incomplete sentence.
Author Response
Point 1: Garcia-Pardo et al have written a brief overview on the potential for future vaccine development in NSCLC. They nicely compile the past attempts and outcomes from multiple clinical trials. This is a good, but brief review on this topic.
Response 1: We thank the reviewer for the comment. Our aim was to perform a comprehensive scoping review and we tried to remain clear, concise and succinct not overburdening the reader. Thanks again for the comment.
Point 2: Major Points
Increased discussion on genomic stability in NSCLC and its role in generating neoantigens
Response 3: Thanks for the suggestion. We added a new paragraph discussing the genomic instability and its role in generating neoantigensm and added references.
Point 3: Minor Points
Line 66 appears to have an incomplete sentence
Response 3: Thanks for the comment. This has been reviewed. Thank you for your review.